# *Toxoplasma gondii* Proteasome Subunit Alpha Type 1 with Chitosan: A Promising Alternative to Traditional Adjuvant

**DOI:** 10.3390/pharmaceutics13050752

**Published:** 2021-05-19

**Authors:** Zhengqing Yu, Wenxi Ding, Muhammad Tahir Aleem, Junzhi Su, Junlong Liu, Jianxun Luo, Ruofeng Yan, Lixin Xu, Xiaokai Song, Xiangrui Li

**Affiliations:** 1MOE Joint International Research Laboratory of Animal Health and Food Safety, College of Veterinary Medicine, Nanjing Agricultural University, Nanjing 210000, China; 2018207044@njau.edu.cn (Z.Y.); 2018807149@njau.edu.cn (W.D.); 2018207076@njau.edu.cn (M.T.A.); 2019807150@njau.edu.cn (J.S.); yanruofeng@njau.edu.cn (R.Y.); xulixin@njau.edu.cn (L.X.); songxiaokai@njau.edu.cn (X.S.); 2State Key Laboratory of Veterinary Etiological Biology, Key Laboratory of Veterinary Parasitology of Gansu Province, Lanzhou Veterinary Research Institute, Chinese Academy of Agricultural Sciences, Lanzhou 730046, China; liujunlong@caas.cn (J.L.); luojianxun@caas.cn (J.L.)

**Keywords:** *Toxoplasma gondii*, proteasome subunit alpha type 1, incomplete Freund’s adjuvant, chitosan, immune protection, mouse, chicken

## Abstract

As an important zoonotic protozoan, *Toxoplasma gondii* (*T. gondii*) has spread around the world, leading to infections in one-third of the population. There is still no effective vaccine or medicine against *T. gondii*, and recombinant antigens entrapped within nanospheres have benefits over traditional vaccines. In the present study, we first expressed and purified *T. gondii* proteasome subunit alpha type 1 (TgPSA1), then encapsulated the recombinant TgPSA1 (rTgPSA1) in chitosan nanospheres (CS nanospheres, rTgPSA1/CS nanospheres) and incomplete Freund’s adjuvant (IFA, rTgPSA1/IFA emulsion). Antigens entrapped in CS nanospheres reached an encapsulation efficiency of 67.39%, and rTgPSA1/CS nanospheres showed a more stable release profile compared to rTgPSA1/IFA emulsion in vitro. In vivo, Th1-biased cellular and humoral immune responses were induced in mice and chickens immunized with rTgPSA1/CS nanospheres and rTgPSA1/IFA emulsion, accompanied by promoted production of antibodies, IFN-γ, IL-4, and IL-17, and modulated production of IL-10. Immunization with rTgPSA1/CS nanospheres and rTgPSA1/IFA emulsion conferred significant protection, with prolonged survival time in mice and significantly decreased parasite burden in chickens. Furthermore, our results also indicate that rTgPSA1/CS nanospheres could be used as a substitute for rTgPSA1/IFA emulsion, with the optimal administration route being intramuscular in mass vaccination. Collectively, the results of this study indicate that rTgPSA1/CS nanospheres represent a promising vaccine to protect animals against acute toxoplasmosis.

## 1. Introduction

*Toxoplasma gondii* (*T. gondii*), which can cause toxoplasmosis, is a highly neglected parasite that threatens the health of animals and humans around the world [1]. As the main form of transmission, sexual reproduction of *T. gondii* occurs in felids, resulting in a wide distribution of oocysts in the environment. *T. gondii* infection can be acquired through ingestion of uncooked food or contaminated water, organ transplantation, blood transfusion, and even maternally by vertical transmission [2,3,4]. Toxoplasmosis can cause abortion in pregnant women, while it is normally asymptomatic in healthy individuals [5]. Effective drug treatments with few side effects are still not been available; thus, efficient vaccines with high biosafety are urgently needed. Anti-*T. gondii* vaccine development strategies are mainly focused on live attenuated vaccines [6], multi-epitope vaccines [7], DNA vaccines [8], subunit vaccines [9], and live vector-based vaccines [10]. Only one live attenuated vaccine, Ovilis™ Toxovax, has been approved to prevent abortion in sheep [11], but a recent document published by the Food Standards Agency questioned its protectiveness against tissue cyst formation [12].

A successful vaccine should be equipped with properties including high safety, long-term immunogenicity, stability, and low cost [13]. Actually, the major obstacle to effective vaccine development is epitope selection [14]. The selected epitope must be highly immunogenic and less prone to causing allergic or autoimmune reactions [15]. Currently, numerous antigens, such as microneme proteins [16], rhoptry proteins [17], dense granule proteins [18], and surface antigens [19], have been reported in the development of subunit vaccines, but a suitable vaccine for inhibiting the replication of this parasite in humans is still unavailable [20,21]. In eukaryotic cells, most of the intracellular proteins are degraded by lysosomes and proteasomes [22]. As an efficient way to degrade short-lived and structurally aberrant proteins [23,24], the ubiquitin-proteasome system (UPS) plays a crucial role in regulating many cellular processes; these process include cell proliferation, cell apoptosis, cell cycle progress [25], signal transduction [26], immune responses [27], and even normal metabolic activity [28]. The key role of protease activity in cellular homeostasis has been illustrated in a large number of eukaryotes as well as protozoan parasites [29]. Studies have shown that the proteasome plays a leading role in the development into a pathogenic stage [30,31]. The intracellular localization and enzymatic activity of the *T. gondii* proteasome have been examined [32], and the proteasome predominates in the cytosol, whereas it mostly presents in nucleus and cytoplasm in eukaryotic cells. To validate the crucial role of the *T. gondii* proteasome, free tachyzoites were incubated with proteasome inhibitors (lactacystin), and the results showed that the proteasome inhibitors could block DNA synthesis and the duplication of intracellular parasites [33,34]. Thus, compounds that specifically target protease activity may constitute a basis for a rational anti-*T. gondii* therapeutic strategy. Of note, the immune protection of the *T. gondii* proteasome in a vaccine formulation against toxoplasmosis has not been investigated.

Purified proteins generally show weak immunogenic activity and require adjuvant to enhance their immunogenicity [35]. Freund’s adjuvant is commonly used and is considered as one of the most effective adjuvants [36]. However, complete Freund’s adjuvant (CFA) is unsuitable for humans due to its serious local reactions and toxicity, while incomplete Freund’s adjuvant (IFA) is characterized by lower toxicity and is only allowed in veterinary vaccines [37,38]. With the introduction of nanotechnology in the medical field, there have been efforts to enhance the immunogenicity of antigens in vaccine delivery systems [39]. Among the many types of nanoparticles, chitosan (CS) has attracted particular interest. Obtained by full or partial deacetylation of chitin, CS is characterized by biocompatibility, biodegradability, relatively no side effects, and nontoxicity [40]. Furthermore, CS has been demonstrated to be safe in dietary applications in some trials [41], and it has been approved for wound dressings by the US Food and Drug Administration (FDA) [42]. Widely used in therapeutic strategies against pathogenic protozoa and parasites [43], CS has proven to be a promising system for developing vaccines and drugs against *T. gondii* [44,45]. So far, CS has been used in peptide-based [7], split [46], subunit [47], and DNA vaccines [48]. As an effective and nontoxic delivery system, CS has been extensively used in vaccine development against toxoplasmosis [49].

In the present study, we present a new attempt to combine recombinant *T. gondii* proteasome subunit alpha type 1 (rTgPSA1) with chitosan nanospheres (CS nanospheres) to develop an anti-*T. gondii* vaccine. Furthermore, we evaluate CS nanospheres along with IFA emulsion for the immunization of mice and chickens using a low-volume, double-dose protocol through the intramuscular route. This study aims to contribute information on a subunit vaccine against toxoplasmosis in mice and chickens delivered by CS microspheres.

## 2. Materials and Methods

### 2.1. Mice, Rats, Poultry, Cell Lines, and T. gondii RH Strain

Specific pathogen-free (SPF) BALB/c mice weighing 18–22 g were bought from the Center of Comparative Medicine, Yangzhou University, China, and housed in strictly sterilized rooms (specific pathogen-free). Also acquired from the Center of Comparative Medicine, SPF Sprague Dawley (SD) rats weighing 200–220 g were reared in a specific pathogen-free environment. Commercially available Hy-Line layer chickens (newborn) were purchased from Shuangli Hatchery, Nantong, China. The chickens were reared in *Toxoplasma*-free conditions. All animals had free access to food and water. All animal procedures were approved by the Animal Ethics committee of responsible authority from the College of Veterinary Medicine, Nanjing Agricultural University, China. The approval ID was PZ2019080 (for mice and rats, approval date: 7 September 2019) and PTA2020006 (for chickens, approval date: 4 June 2020).

Human foreskin fibroblast (HFF) cells were purchased from the Institute of Cell Biology, Chinese Academy Sciences, Shanghai, China, and maintained at 37 °C in a 5% CO_2_ atmosphere in Dulbecco’s Modified Eagle Medium (DMEM; Gibco, Carlsbad, CA, USA) containing 2% fetal bovine serum (FBS; Gibco, Carlsbad, CA, USA) and 1% double antibiotics (Gibco, Carlsbad, CA, USA).

Violent *T. gondii* strains RH (Type I) were preserved at the MOE Joint International Research Laboratory of Animal Health and Food Safety, College of Veterinary Medicine, Nanjing Agricultural University, Nanjing, China. Tachyzoites of *T. gondii* were amplified through mice following methods described previously [50]. For preparation of chicken immunization doses, tachyzoites were harvested from HFF cells challenged with *T. gondii* referenced in a previous report [51].

### 2.2. Cloning and Molecular Characterization of TgPSA1

A Total RNA Extraction Kit (Omega Bio-tek, Norcross, GA, USA) and reverse transcription kit (Takara Biotechnology, Dalian, China) were used to obtain the total cDNA from the tachyzoites of *T. gondii*. Primers were designed according to the conserved domain sequences (CDS) of the TgPSA1 gene (GenBank: AAYL02000522). Synthesized by Tsingke Biological Technology (Nanjing, China), sense and antisense primers 5′-GCTGATATCGGATCC GAATTC ATGCTGGCTGGCGGCC-3′ and 5′-CTCGAGTGCGGCCGC AAGCTT TTAGTCAGCGCGGAGGTCCA-3′ were used to amplify the complete open reading frame (ORF) of TgPSA1 using the cDNA as a template. The underlined sequences respectively represent the restriction endonuclease of *Eco*RI and *Hind*III (Takara Biotechnology, Dalian, China). To amplify the TgPSA1 gene, *Ex Taq* DNA polymerase (Takara Biotechnology, Dalian, China) was used. The PCR was conducted with the following conditions in an amplifier (Thermo Scientific, Waltham, MA, USA): preheating at 95 °C for 5 min followed by 35 cycles with amplification at 95 °C for 30 s, 64 °C annealing for 30 s, 72 °C extension for 60 s, and final extension at 72 °C for 5 min. Then, the PCR amplicons were subcloned into the pET32a linear vector (Takara Biotechnology, Dalian, China) using the One Step Cloning Kit (Vazyme Biotech, Nanjing, China). The recombinant plasmid was then transferred to *Escherichia coli* (*E. coli*) BL21 (DE3, Invitrogen Biotechnology, Shanghai, China) and cultured in Luria Bertani (LB) medium containing 100 μg/mL ampicillin. The recombinant vector was extracted by using a Plasmid Mini Kit (Omega Bio-tek, Norcross, GA, USA), and then sequenced by an ABI PRISM™ 3730 XL DNA Analyzer (Applied Biosystems, Waltham, MA, USA).

### 2.3. Preparation of Purified Recombinant TgPSA1

To produce the recombinant protein of the TgPSA1 gene, *E. coli* BL21 (DE3) containing pET32a/TgPSA1 vector was grown in LB medium containing 100 μg/mL ampicillin (37 °C with shaking at 180 rpm) until the OD600 was around 0.5. Then, isopropyl-β-D-thiogalactopyranoside (IPTG; Sigma-Aldrich, St. Louis, MO, USA) was added for a final concentration of 1 mM. After 4 h of incubation, the recombinant protein was then purified by a chelating column (Ni-NTA, GE Healthcare, Piscataway, NJ, USA) following the manufacturer’s instructions. The endotoxin was removed from the purified protein (ToxinEraser™ Endotoxin Removal Kit, GeneScript, Piscataway, NJ, USA), the level of endotoxin was determined (ToxinSensor™ Chromogenic LAL Endotoxin Assay Kit, GeneScript, Piscataway, NJ, USA), and the endotoxin was quantified by a Pierce™ BCA Protein Assay Kit (Thermo Scientific, Waltham, MA, USA), analyzed by 12% sodium dodecyl sulfate-polyacrylamide gel electrophoresis (SDS-PAGE), and visualized by Coomassie brilliant blue dye (Solarbio, Beijing, China).

### 2.4. Immunoblot Analysis of Recombinant and Native TgPSA1

To obtain the polyclonal antibodies against rTgPSA1, two SD rats were first immunized with 200 μg of purified protein mixed with oil adjuvant (complete Freund’s adjuvant; Sigma-Aldrich, St. Louis, MO, USA). After two weeks, quartic immunizations were conducted using 200 μg of purified protein mixed with IFA (Sigma-Aldrich, St. Louis, MO, USA) at one-week intervals. All injections were performed subcutaneously in the back skin through multiple injections. One week after the last immunization, blood was harvested at the eye socket and sera containing polyclonal antibodies were collected. Blank sera as negative control were also collected from rats and kept in the same environment. To obtain the antibodies against *T. gondii*, two SD rats were challenged with 10^3^ tachyzoites of *T. gondii* RH strain intraperitoneally. After four weeks, sera containing antibodies against *T. gondii* were harvested. All sera were kept at −20 °C until use.

To obtain the lysates of *T. gondii* tachyzoites, a previously described procedure was conducted [52]. Purified rTgPSA1 and *T. gondii* lysates were resolved by 12% SDS-PAGE and transferred onto polyvinylidene difluoride (PVDF) membranes (Millipore, Billerica, MA, USA), which had been cut into strips. The membranes were then incubated with Tris-buffered saline (TBS) containing 5% skim milk (*w*/*v*) and 0.5% Tween 20 (*v*/*v*) for 2 h at 37 °C with constant shaking. After being washed three times in TBS with 0.5% Tween 20 (*v*/*v*, TBST), the membranes were subsequently probed with sera (1:100) at 37 °C for 2 h with constant shaking. After being washed with TBST three times, the membranes were incubated with horseradish peroxidase (HRP) labeled anti-rat IgG (1:5000; eBioscience, San Diego, CA, USA) for 1 h at 37 °C with constant shaking. Finally, the membranes were washed three times in TBST and detected by freshly prepared 3,3′-diaminobenzidine (DAB; Sigma-Aldrich, St. Louis, MO, USA) for 5 min.

### 2.5. Nanosphere Formulation (rTgPSA1/CS) and Physical Characterization

The ionic gelation technique was carried out to prepare nanospheres as described previously with minor modifications [53]. To form a 2 mg/mL CS solution (pH 5.0), the CS (MW 50–190 kDa; Sigma-Aldrich, St. Louis, MO, USA) was dissolved in an aqueous solution of acetic acid (1%, *v/v*) on a magnetic stirrer (400 rpm) overnight at room temperature; the pH value was then adjusted by using 2 M NaOH solution. For the 2 mg/mL sodium tripolyphosphate (TPP) solution, TPP was dissolved in double-distilled water and passed through a 0.22 μm filtering membrane (Millipore, Billerica, MA, USA).

To prepare rTgPSA1/CS nanospheres, a magnetic stirrer was put in an incubator, in which the temperature was 30 °C, and the disturbance of cold flow was avoided. Then, 20 mL CS solution in a container (preheated in a water bath at 37 °C) was placed on a magnetic stirrer stirring at 500 rpm; 4 mL TPP solution was then cautiously added with a plastic pipette tip. After incubation for 10 min, 4 mg recombinant protein was cautiously dripped. Tip sonication (Scientz Biotechnology, Ningbo, China) was then performed in continuous mode for 5 s at 3 s intervals (10 min total) under output power of 50 W. During the ultrasonic treatment process, the mixed solution was cooled in an ice bath. Before centrifugation at 40,000 rpm (15 min at 4 °C), the total volume of the resulting solution was recorded. The supernatant of the solution after centrifugation was collected and its volume measured, and the obtained nanospheres were rinsed and resuspended in double-distilled water. Before freezing for at least 2 h at −80 °C, the obtained nanospheres were passed through a 0.22 μm filtering membrane. The frozen nanospheres were quickly transferred to a vacuum freeze-dryer (Labconco, Kansas City, MO, USA) until completely freeze-dried.

To analyze the encapsulation efficiency, the non-bound protein in the supernatant was quantified by a Pierce™ BCA Protein Assay Kit, and the total amount of recombinant protein in the supernatant could be calculated. The encapsulation efficiency can be evaluated by Formula (1):(1)$$\mathrm{Encapsulation}\;\mathrm{efficiency}\;\left(\%\right)=\frac{\mathrm{Total}\;\mathrm{protein}-\mathrm{Free}\;\mathrm{protein}}{\mathrm{Total}\;\mathrm{protein}}\times 100\%$$

To characterize the features of rTgPSA1/CS nanospheres, the nanospheres were sent to Nanjing Agriculture University for scanning electron microscope (SEM) observation by a Hitachi SU8010 (Tokyo, Japan). Using ImageJ software, version 1.8 (NIH, Bethesda, MD, USA), the average diameter was determined by measuring five arbitrary nanospheres.

### 2.6. Release Characteristics In Vitro

As described previously [54], the release characteristics of rTgPSA1/CS nanospheres in vitro were investigated in PBS (pH 7.4) with minor changes. To form the rTgPSA1/IFA emulsion (water-in-oil emulsion), an equal volume of IFA was added to the recombinant protein solution. This was placed in an ice-water mixture, then tip sonication was performed in continuous mode for 3 s at 3 s intervals (4 min total) under output power of 20 W (with 40% amplitude). After vertexing thoroughly, tip sonication was repeated. The rTgPSA1/CS nanospheres and rTgPSA1/IFA emulsion were dispersed at 37 °C on a magnetic stirrer stirring at 180 rpm. At 12-h intervals, supernatant samples were taken out and centrifuged for 1 min at 10,000 rpm. The aqueous solution was stored at −20 °C and the concentration of protein was quantified by using a Pierce™ BCA Protein Assay Kit. The rTgPSA1/CS nanospheres that settled at the bottom and the emulsion that floated on the aqueous solution were collected and resuspended in the original tubes. The cumulative release percentage was calculated by Formula (2). Before calculating, the total volume of rTgPSA1/CS nanospheres and rTgPSA1/IFA emulsion in the supernatant was measured.
(2)$$\mathrm{Cumulative}\;\mathrm{release}\;\left(\%\right)=\frac{\mathrm{Total}\;\mathrm{volume}\times \;\mathrm{Protein}\;\mathrm{concentration}}{\mathrm{Total}\;\mathrm{amount}\;\mathrm{of}\;\mathrm{loaded}\;\mathrm{protein}}\times 100\%$$

### 2.7. Vaccination and Challenging Schedules in Mice

BALB/c mice were randomly divided into five groups (10 mice/group) and immunized intramuscularly in the leg with 100 μg of recombinant protein at different sites. The immunized mice were injected with PBS, pET32a vector protein, rTgPSA1, rTgPSA1/IFA, and rTgPSA1/CS (Table 1). The rTgPSA1/IFA emulsion was prepared following the method described in Section 2.6. Two weeks after the first immunization, a booster immunization was performed in the same way.

Two weeks after the second immunization, the immune protection of immunized mice was evaluated by a method described previously [55]. Briefly, all mice in every group were challenged with a lethal dose (200 tachyzoites) of *T. gondii* RH strain intraperitoneally, and the mortality status of each mouse was observed daily.

### 2.8. Antibody and Cytokine Assays in Mice

At weeks zero, two, and four, serum from each mouse was collected from the eye socket before immunization and challenge. All harvested sera were kept at −20 °C until use.

To determine the titers of antibodies, standard enzyme-linked immunosorbent assay (ELISA), as described previously, was conducted with minor modifications [56]. Briefly, the wells of the 96-well plate (Costar, Cambridge, MA, USA) were coated with 100 μL (100 μg/mL) rTgPSA1 in 50 mM carbonate buffer (pH 9.6) for 12 h at 4 °C. After being washed three times in PBST (PBS containing 0.05% Tween 20), the nonspecific binding sites were blocked with PBST containing 3% bovine serum albumin (BSA) at 37 °C for 1 h. After being rinsed for 5 min in PBST, the wells were then incubated with sera (diluted 1:100) from immunized mice at 37 °C for 1 h. After subsequent washing in PBST, each well was then incubated with HRP-bound anti-mouse IgG, IgG1, or IgG2a (1:5000; eBioscience, San Diego, CA, USA). The enzymatic activity was revealed by using substrate solution (3,3′,5,5′-tetramethylbenzidine; Tiangen, Beijing, China), and the reaction was stopped with 2 M newly prepared H_2_SO_4_. A microplate photometer (Thermo Scientific, Waltham, MA, USA) was used to quantify the titers of antibodies at an absorbance of 450 nm. Each group had five replications and each replication was measured once.

To determine the levels of cytokines in the sera, commercially available ELISA kits (Senbeijia, Nanjing, China) based on the double-antibody sandwiched method were conducted. According to known concentrations of recombinant mouse cytokines, the concentrations of interferon-gamma (IFN-γ), and interleukin (IL)-4, IL-10, and IL-17 were measured. Each group was in five replications and each replication was measured once.

### 2.9. Determination of Optimal Administration Route

BALB/c mice were randomly divided into 20 groups (5 mice/group). The immunization routes were intraoral, intranasal, intramuscular, and subcutaneous, and each route included blank (equal volume of PBS), control (100 μg pET32a vector protein), rTgPSA1 (100 μg rTgPSA1), rTgPSA1/IFA (rTgPSA1/IFA emulsion containing 100 μg rTgPSA1), and rTgPSA1/CS (rTgPSA1/CS nanospheres containing 100 μg rTgPSA1) groups. For the intraoral group, immunization was administered by the intragastric method. For the intranasal group, each mouse was placed in an anesthetic apparatus and induced with 2% isoflurane with oxygen flow at 1 L/min, and maintained at 1.5%. A 10 μL drip containing 50 μg recombinant protein was then given to each nostril (20 μL/mice). For the intramuscular group, each mouse was immunized in the leg muscles at different sites. For the subcutaneous group, 100 μg recombinant protein was delivered into the back skin, also at different sites. All infected mice were immunized twice at two-week intervals. To evaluate the effect of the administration route on immune protection, all mice were challenged intraperitoneally with a lethal dose of *T. gondii* RH strain (200 tachyzoites) two weeks after the last immunization, and the mortality status of every mouse was then observed daily.

### 2.10. Immunization and Challenging Schedules in Chickens

Newborn Hy-Line layer chickens were raised together under *Toxoplasma*-free conditions. Two weeks later, chickens with similar body weight were randomly divided into six groups (25 chickens/group) and intramuscularly immunized in the leg at different sites. The groups included blank (immunized with PBS or *T. gondii*), control, rTgPSA1, rTgPSA1/IFA, and rTgPSA1/CS groups (Table 2). Two weeks after the first immunization, the second immunization (four weeks old) was conducted in the same way. Based on previous methods [50], immune protection was investigated with minor modifications. All chickens were challenged with 1 × 10^7^ tachyzoites of *T. gondii* RH strain intraperitoneally two weeks after the last immunization (six weeks old). The body weight of 10 chickens was recorded, and the growth coefficient was obtained by Formula (3):(3)$$\mathrm{Coefficient}\;\mathrm{of}\;\mathrm{growth}\;\left(\%\right)=\frac{\mathrm{Final}\;\mathrm{weight}-\mathrm{Initial}\;\mathrm{weight}}{\mathrm{Initial}\;\mathrm{weight}}\times 100\%$$

### 2.11. Antibody and Cytokine Assays in Chickens

At two, four, and six weeks of age, serum from each chicken was collected from the heart before immunization and challenge. All harvested sera were kept at −20 °C until use. The procedure for antibody titers was the same as that described in Section 2.8, whereas the second antibody was changed to HRP-bound anti-chicken IgY (Abcam, Cambridge, MA, USA). There were five replications in each group and each replication was measured once. To determine the cytokine levels in the sera, commercially available ELISA kits (Senbeijia, Nanjing, China) based on a double-antibody sandwiched method were used. According to known concentrations of recombinant chicken cytokines, all concentrations of cytokines (IFN-γ, IL-4, IL-10, and IL-17) were measured. There were five replications in each group and each replication was measured once.

### 2.12. Detection of T Cell Subsets in Spleen Lymphocytes from Chickens

Five animals from each group were sacrificed at two weeks after immunization or challenge (two, four, and six weeks old), and spleen lymphocytes were collected by using a separation solution (TBD, Tianjin, China) as described by Sasai et al. [57]. The spleen lymphocytes were divided into two parts and stained with anti-chicken CD3-FITC, anti-chicken CD4-PE and anti-chicken CD3-FITC, or anti-chicken CD8-PE (eBioscience, San Diego, CA, USA) in the dark for 30 min at 4 °C, according to the manufacturer’s instructions. After being washed three times in PBS, the lymphocytes were run on a flow cytometer (Beckman Coulter Inc., Brea, CA, USA) to calculate the percentage of CD4^+^ and CD8^+^ T cells. Before analysis, a lymphocyte subset was gated. All five replications in each group were measured once.

### 2.13. Detection of T. gondii by Absolute Quantitative PCR (qPCR)

Five chickens in each group were sacrificed at two weeks after challenge (eight weeks old), and the cardiac muscle from the tip of the heart was collected. Then, 30 mg of tissue was minced and transferred in a microcentrifuge tube, and genomic DNA was extracted by using a commercial kit (Omega Bio-tek, Norcross, GA, USA). The extracts were stored at −20 °C until use.

The 529 bp fragments were amplified by PCR using the primers TOX4 and TOX5 as described by Homan et al. [58]. Purified by a gel extraction kit (Omega Bio-tek, Norcross, GA, USA), the amplicons were cloned into linear PMD19T vectors (Takara Biotechnology, Dalian, China) by the TA cloning method. The recombinant vector was sequenced by the ABI PRISM™ 3730 XL DNA Analyzer. Correctly identified sequences were transferred to *E. coli* DH5α (Invitrogen Biotechnology, Shanghai, China), and the *E. coli* were incubated in LB medium containing 100 μg/mL ampicillin for duplications. A Plasmid Mini Kit (Omega Bio-tek, Norcross, GA, USA) was utilized to obtain the cloned vectors. To determine the copy number of obtained vectors, an online tool was used (http://cels.uri.edu/gsc/cndna.html (accessed on 19 May 2021)).

Absolute qPCR for quantification of *T. gondii* 529 bp fragments was conducted following previous methods [58]. The forward (5′-CACAGAAGGGACAGAAGT-3′) and reverse (5′-TCGCCTTCATCTACAGTC-3′) primers were synthesized by Tsingke Biological Technology (Nanjing, China). For qPCR, 1 μL of a known copy number of cloned vectors and 1 μL of genomic DNA were added to the amplification mixture consisting of 10 μL of 2 × ChamQ SYBR qPCR MasterMix (Vazyme, Nanjing, China), 0.4 pmol of each primer, and 0.4 μL of 50 × ROX Reference Dye 2, and the final volume of each reaction was 20 μL. Then, qPCR was conducted on an Applied Biosystems 7500 system (Life Technologies, Carlsbad, CA, USA). Samples were amplified as follows: 95 °C for 30 s, followed by 40 cycles of 95 °C for 10 s and 60 °C for 30 s. Amplification was followed by a melt-curve analysis between 60 and 95 °C with an increment of 0.05 °C/s. The fluorescence of each sample was collected at the extension step. Before further processing, the melting curves were ensured with one uniform peak at the expected temperature. Each cloned vector and genomic DNA sample was quantified with three biological replicates.

### 2.14. Statistical Analysis

All statistical analysis was performed by GraphPad 6.0 software (GraphPad Prism, San Diego, CA, USA) and SPSS 25.0 software (SPSS Inc., Chicago, IL, USA). The data were shown as mean ± standard deviation (SD), and differences between groups were considered significant at *p* < 0.05 using one-way ANOVA. Flow cytometric analysis was conducted by CytExpert software (version 2.3, Beckman Coulter Inc., Brea, CA, USA). Dunnett’s test was used for comparison between experimental and control groups. Based on the homogeneity of variance, values between the rTgPSA1/IFA and rTgPSA1/CS groups were compared with the independent *t*-test. When variance was uneven, Welch’s correction was performed. Survival percentages were calculated by the Kaplan–Meier survival test, and the log-rank model was used for comparison.

## 3. Results

### 3.1. Cloning, Expression, and Purification of rTgPSA1

The sequence analysis showed that the pET32a/TgPSA1 plasmid was correctly inserted. The ORF of TgPSA1 was 738 bp, which encoded a protein of 246 amino acids with an expected molecular weight of 27.2 kDa. According to the instructions, the recombinant protein expressed by the pET32a vector involved his-tag protein (17.5 kDa). Therefore, the molecular weight of the recombinant protein was 44.7 kDa in theory. After removing the endotoxin from purified rTgPSA1, the endotoxin level of purified rTgPSA1 fell to 0.1 EU/mL. As shown in Figure 1a, SDS-PAGE analysis revealed that the molecular weight of rTgPSA1 was approximately 44 kDa, which matched the theoretical value.

### 3.2. Immunoblot Analysis of Recombinant and Native TgPSA1

Immunoblot analysis indicated that rTgPSA1 could be detected by the mouse anti-*T. gondii* sera (Figure 1b), while native TgPSA1 could be recognized by anti-rTgPSA1 polyclonal antibodies (Figure 1c). Compared with the calculated value, the molecular weight of the native TgPSA1 protein was slightly lower. All of these results show that the antigenicity of the recombinant TgPSA1 was satisfactory, and the expressed protein could participate in activating host immunity.

### 3.3. Physical Characterization and Release Characteristics of rTgPSA1/CS Nanospheres

CS encapsulated rTgPSA1 nanospheres were prepared by the ionic gelation technique, and the encapsulation efficiency was analyzed. The concentration of recombinant protein solution used in the preparation reached 1.0 mg/mL, and the encapsulation efficiency reached 67.39%. After encapsulation of rTgPSA1 protein in CS polymer, the morphology of nanospheres was observed under SEM. The average diameter of encapsulated rTgPSA1/CS nanospheres was around 100 nm (Figure 2).

Based on BCA assay, the release characteristics of rTgPSA1/CS nanospheres and rTgPSA1/IFA emulsion were further evaluated. The encapsulated nanospheres showed a steadier release profile (Figure 3); only 10.150 ± 1.917% of the antigens were released within the first day, and 1.18–6.00% were slowly released daily over the following 10 days. After the 12th day, the release profile became flat, finally reaching 76.236 ± 2.870%. Meanwhile, the rTgPSA1/IFA emulsion showed an obvious burst release during the first two days, and reached 63.060 ± 2.612% on the third day. After that, a steadier release profile was detected, and the cumulative release was 90.436 ± 2.408% for 16 days.

### 3.4. Antibody and Cytokine Production in Mice

According to the standard ELISA procedure, the titers of total IgG and isotypes IgG1 and IgG2 in the sera collected from mice were investigated. As shown in Figure 4a, significantly increased levels of total IgG antibodies were found in rTgPSA1, rTgPSA1/IFA, and rTgPSA1/CS groups after the first (week two) and second (week four) immunizations, and mice immunized with rTgPSA1/IFA generated the highest level (*p* < 0.01) of total IgG after the second immunization (week four). Furthermore, the sera separated from animals also revealed higher titers of IgG1 (Figure 4b) and IgG2a (Figure 4c) after the first and second immunizations. Compared with the blank and control groups, animals immunized with rTgPSA1, rTgPSA1/IFA, or rTgPSA1/CS generated higher levels (*p* < 0.001) of isotype IgG1 after the first (week two) and second (week four) immunizations. As for isotype IgG2a, remarkable increases (*p* < 0.001) were also observed in the rTgPSA1, rTgPSA1/IFA, and rTgPSA1/CS groups after the first (week two) and second (week four) immunizations. Furthermore, the rTgPSA1/IFA group secreted higher levels (*p* < 0.05) of isotype IgG2a than the rTgPSA1/CS group.

Strictly according to the instructions, the sera collected from animals after the second immunization (week four) were tested to determine the levels of IFN-γ, IL-4, IL-10, and IL-17. As demonstrated in Figure 5a,b, obviously higher levels of IFN-γ and IL-4 were detected in the rTgPSA1, rTgPSA1/IFA, and rTgPSA1/CS groups. However, all animals from immunized groups generated equivalent IL-10 (*p* > 0.05; Figure 5c). As for IL-17, shown in Figure 5d, the secretions in animals immunized with rTgPSA1 and rTgPSA1/IFA were enhanced, while those in the rTgPSA1/CS group remained unchanged (*p* > 0.05).

### 3.5. Immune Protection against Acute Toxoplasmosis

To investigate the immune protection against acute *T. gondii* infection, all animals were intraperitoneally immunized with 200 newly prepared tachyzoites (*T. gondii* RH strain) two weeks after the second vaccination. As illustrated in Figure 6, all immunized animals died within 17 days. Significant survival time was observed in mice vaccinated with rTgPSA1 (12.200 ± 0.490 days, *p* < 0.01), rTgPSA1/IFA (13.700 ± 0.700 days, *p* < 0.001), and rTgPSA1/CS (11.400 ± 0.476 days, *p* < 0.001) compared with the blank group (9.300 ± 0.335 days). Longer survival time was also observed in the rTgPSA1 (*p* < 0.01), rTgPSA1/IFA (*p* < 0.001) and rTgPSA1/CS (*p* < 0.001) groups based on the control group (9.100 ± 0.379 days). No significant activity (*p* > 0.05) was observed in animals immunized with rTgPSA1/IFA and rTgPSA1/CS.

### 3.6. Comparison of Administration Routes

In order to evaluate the ability of rTgPSA1/CS nanospheres to induce immune protection via the four pathways, the survival time was recorded in mice challenged with 200 newly prepared tachyzoites (*T. gondii* RH strain). As illustrated in Figure 7, immunization with rTgPSA1/IFA emulsion showed longer survival time through the four investigated routes, while the rTgPSA1/CS nanospheres prolonged survival time only through the intranasal, intramuscular, and subcutaneous routes. With the intraoral route, immunization with rTgPSA1/IFA emulsion significantly increased the survival duration based on the blank and control groups, while with the intranasal, intramuscular, and subcutaneous routes, both rTgPSA1/IFA emulsion and rTgPSA1/CS nanospheres significantly prolonged the survival time of mice based on the blank and control groups. No significant difference (*p* > 0.05) between the rTgPSA1/IFA and rTgPSA1/CS groups was observed in the four investigated routes, and no notable difference (*p* > 0.05) was evaluated between the blank and control groups using the same administration route.

### 3.7. Antibody and Cytokine Production in Chickens

According to the ELISA procedure mentioned above, the titers of total IgY in sera collected from chickens were investigated. As shown in Figure 8, statistically higher IgY titers were observed in the rTgPSA1, rTgPSA1/IFA, and rTgPSA1/CS groups at four and six weeks old. No obvious difference (*p* > 0.05) was detected between the rTgPSA1/IFA and rTgPSA1/CS groups at any point in time, and the blank group was consistent (*p* > 0.05) with the control group.

Using the ELISA kits constructed with the double antibody sandwich method, the serum separated from each chicken two weeks after the second immunization (six weeks old) was conducted for levels of IFN-γ, IL-4, IL-10, and IL-17.

As demonstrated in Figure 9a, IFN-γ secretion was promoted (*p* < 0.001) in the rTgPSA1, rTgPSA1/IFA, and rTgPSA1/CS groups compared with the blank and control groups. As for cytokine IL-4, illustrated in Figure 9b, significant enhancements were evaluated in the sera separated from the rTgPSA1 (*p* < 0.001), rTgPSA1/IFA (*p* < 0.001), and rTgPSA1/CS (*p* < 0.05) groups compared with the blank group. When compared with the control group, statistical increases were also found in the rTgPSA1 (*p* < 0.001), rTgPSA1/IFA (*p* < 0.01) and rTgPSA1/CS (*p* < 0.05) groups.

As illustrated in Figure 9c, the rTgPSA1/IFA emulsion significantly (*p* < 0.001) promoted higher levels of IL-10 compared to the blank and control groups. High levels were also detected in the chickens immunized with rTgPSA1/CS nanospheres, which showed a significant difference (*p* < 0.05) compared with the control group. Noticeably, chickens in the rTgPSA1/IFA group generated higher levels of IL-10 (*p* < 0.05) than those in the rTgPSA1/CS group. Regarding the secretion of cytokine IL-17 (Figure 9d), only chickens immunized with rTgPSA1/IFA emulsion showed higher levels. In addition, the blank group was consistent with the control group (*p* > 0.05) in four investigated cytokines.

### 3.8. Analysis of Cellular Immune Response in Spleen Lymphocytes Separated from Chickens

To determine the percentage of CD4^+^ T and CD8^+^ T cells after immunization, five chickens from each group were sacrificed and spleen lymphocytes were collected. As illustrated in Figure 10a, the percentage of CD4^+^ T cells in the rTgPSA1, rTgPSA1/IFA, and rTgPSA1/CS groups was significantly increased at four and six weeks of age. The independent *t*-test revealed that the percentage in the rTgPSA1/CS group was significantly higher (*p* > 0.05) than that in the rTgPSA1/IFA group at 4 weeks of age. As for CD8^+^ T cells (Figure 10b), increases were also detected in the rTgPSA1, rTgPSA1/IFA, and rTgPSA1/CS groups after the first (four weeks old) and second (six weeks old) immunizations. The blank group was statistically similar (*p* > 0.05) to the control group in the investigated T lymphocytes.

### 3.9. Growth Coefficient and Parasite Burden of Chickens

In order to evaluate the coefficient of growth, the weight of chickens was measured. As shown in Figure 11, the growth rate was significantly suppressed (*p* < 0.001) in all challenged animals in the six-to-eight-week-old period compared to the blank group without *T. gondii* challenge. Animals from the rTgPSA1/IFA and rTgPSA1/CS groups were observed to have a higher coefficient of growth (*p* < 0.001) than those from the blank (*T. gondii*) and control groups, while the rTgPSA1 group showed a higher growth rate (*p* < 0.05) than the control group. The coefficient of growth for the control group was similar (*p* > 0.05) to that of the blank group (*T. gondii*), and the growth of chickens in the rTgPSA1/IFA group was in line with that of the rTgPSA1/CS group (*p* > 0.05) according to the independent *t*-test.

In order to carry out a more accurate analysis of the parasite burden, absolute qPCR was conducted to detect the 529 bp fragments of *T. gondii* in cardiac muscles from chickens at 8 weeks of age. As shown in Figure 12, the parasite burden was statistically inhibited in the rTgPSA1, rTgPSA1/IFA, and rTgPSA1/CS groups compared to the blank and control groups. In addition, no differences (*p* > 0.05) were obtained in copy numbers of 529 bp fragments between the rTgPSA1/IFA and rTgPSA1/CS groups.

## 4. Discussion

Currently, various adjuvants are used to develop anti-*T. gondii* vaccines, and the use of nano- and microspheres has gained popularity [59]. Approved by the US Food and Drug Administration [60], poly (d, l-lactic-co-glycolic acid) (PLGA) has been shown to be an effective adjuvant to *T. gondii* vaccines [61,62]. However, during the preparation of PLGA nanospheres, carcinogenic dichloromethane cannot be avoided. In addition, various nano-adjuvants have been developed for *T. gondii* vaccines [63,64]. To some degree, all of these reported nano-adjuvants can protect antigens from enzymatic degradation and maintain a stable release of antigens, leading to enhanced immunogenicity. As a newly arising nano-adjuvant, CS can not only effectively induce cell and humoral immune responses [65], but also stimulate innate and adaptive immunity. Moreover, CS possesses a slow-release effect, which could extend the immunogenicity of entrapped antigens and regulate macrophages and dendritic cells (DCs) in the host [66,67,68].

Different methods for developing CS nanospheres can affect the encapsulation efficiency, mainly including emulsion cross-linking [7], solvent evaporation [69], spray-drying [70], emulsion precipitation [71], and ionic gelation [53]. To avoid toxic substances, a modified ionic gelation technique was utilized in the present study, and the encapsulation efficiency only reached 67.39%. Further research should optimize the nanosphere protocol to improve the encapsulation efficiency. As a critical attribute, sphere size plays an important role in inducing efficient antigenic uptake by antigen-presenting cells (APCs) and making them accessible to the appropriate DC subset [72]. In this study, we formulated recombinant TgPSA1 into CS as an adjuvant and vaccine delivery system. The rTgPSA1/CS nanospheres in our study were spherical in shape and had a uniform diameter of approximately 100 nm, according to the SEM images. Using the same technique, Temsahy et al. [46] produced CS nanospheres with an average diameter of 211.896 ± 67.5812 nm. Such differences could be due to different antigens and methods. Furthermore, negatively charged CS nanospheres are composed of a cationic polymer that can adhere to cell surfaces, leading to a long residence time and continuous antigen release [73]. Thus, the release profile of rTgPSA1/CS nanospheres was investigated; the curve showed an initially slow release, which was fastest on approximately the 6th day, and the release profile tended to increase gently after the 12th day. Similar release profiles were also observed for CS nanospheres loaded with multiple antigenic epitopes from *T. gondii* GRA10 [7].

In resisting the replication of this intracellular parasite, humoral responses and antibodies play important roles [74,75]. In mice, it has been reported that elevated IgG2a is an indicator for a Th1 type immune response, while elevated IgG1 indicates a Th2 type response [55]. In our study, significantly increased total IgG, IgG1, IgG2a, and IgY secretion was observed, and higher titers could be induced after booster immunization. With higher titers of IgG2a than IgG1, rTgPSA1/CS nanospheres and rTgPSA1/IFA emulsion induced a Th1-biased immune response in mice. In addition, rTgPSA1/IFA emulsion generated significantly higher total IgG and IgG2a titers than rTgPSA1/CS nanospheres in mice while, in chickens, both rTgPSA1/CS nanospheres and rTgPSA1/IFA emulsion generated significantly higher IgY. Our data lend credence to the idea that rTgPSA1/CS nanospheres could induce a Th1-oriented immune response and could be a substitute for IFA to trigger specific antibodies in chickens. Similar observations have also been reported for many CS-based vaccine formulations against *T. gondii* [7,48,76].

Secreted by natural killer (NK) and T cells, cytokine IFN-γ plays an essential role in resisting *T. gondii* [77,78] and in mediating innate and adaptive immune responses [79]. In the present study, both rTgPSA1/CS nanospheres and rTgPSA1/IFA emulsion induced statistically higher IFN-γ in mice and chickens, and no statistical difference was observed between the two vaccines. Furthermore, excessive secretion of IFN-γ can cause severe inflammation, leading to the death of sick animals [80]. It has been reported that Th2 immunity-related IL-4 and IL-10 are resistant to IFN-γ [81]. Thus, we investigated IL-4 and IL-10 and found that rTgPSA1/CS nanospheres and rTgPSA1/IFA emulsions induced higher IL-4 and lower IL-10 in mice. In chickens, higher IL-4 and IL-10 were triggered by the two vaccines; in addition, rTgPSA1/IFA emulsion induced significantly higher levels of IL-10 than rTgPSA1/CS nanospheres. Th2 immunity could reduce the inflammatory reaction and mortality, and may play a crucial role in prolonging the survival time of animals [82]. To sum up, both rTgPSA1/CS nanospheres and rTgPSA1/IFA emulsion elicited high levels of IFN-γ and other Th2 immunity-related cytokines against *T. gondii* and, in chickens, there are fewer possibilities for rTgPSA1/CS nanospheres to elicit higher IL-10.

Th17 cells can secrete antimicrobial cytokines, including IL-17, IL-21, and IL-22, and play an important role in mediating host protection against parasites [83,84]. IL-17 can enhance the migration of neutrophils to infection sites and participate in immunity to intracellular infections [85]. In the present study, we found that rTgPSA1/IFA emulsion could induce remarkably higher levels of IL-17 in mice and chickens, while IL-17 secretions in animals immunized with rTgPSA1/CS nanospheres remained unchanged. In mice, rTgPSA1/IFA emulsion could induce significantly higher levels of IL-17 compared to rTgPSA1/CS nanospheres. The results indicate that rTgPSA1/IFA emulsion could induce higher IL-17 production than rTgPSA1/CS nanospheres. During the development of anti-*T. gondii* vaccines, the Th17 pathway has received less attention, and further studies are required to elucidate its role in recruiting cells and mediating cytokines.

Composed of CD4^+^ and CD8^+^ T lymphocytes, memory T lymphocytes are crucial in establishing protective immunity against *T. gondii* [78]. CD4^+^ T lymphocytes play a significant role in regulating the immune action against *T. gondii*, while CD8^+^ T lymphocytes are known as the most important effector T cells [86]. Our data show that both CD4^+^ and CD8^+^ T lymphocytes were significantly expanded in response to rTgPSA1/CS nanospheres and rTgPSA1/IFA emulsion in chickens. This observation indicates that both nanospheres and emulsion can induce the activation of CD4^+^ and CD8^+^ T lymphocytes, which may contribute to cytotoxicity against toxoplasmosis. The ability of rTgPSA1/CS nanospheres to stimulate T lymphocyte differentiation was similar to that reported in [62] and, in that study, recombinant *T. gondii* rhoptry 38 and 18 were encapsulated in PLG microparticles.

The survival rate of immunized animals against toxoplasmosis is a convincing way to assess vaccines [87]. Thus, we evaluated the administration route and immune protection in mice. In the present study, we compared the survival time of mice immunized with rTgPSA1/CS nanospheres and rTgPSA1/IFA emulsion via four vaccination pathways. Through the intranasal, intramuscular, and subcutaneous routes, rTgPSA1/CS nanospheres and rTgPSA1/IFA emulsion significantly prolonged the survival time of mice. As the most common administration route in practice, the intraoral route seems to be the easiest and most convenient option. However, it was not effective for the vaccines used in this study compared with the other routes. In a previous study [88], recombinant ubiquitin-conjugated multi-stage antigen segments (Ad-UMAS) derived from different stages of *T. gondii* were synthesized to prevent toxoplasmosis and administered through five different routes, and the results indicated that oral and nasal immunizations significantly prolonged survival time after the challenge. However, in the present study, only rTgPSA1/IFA emulsion could induce limited immune protection; such differences may be caused by different types of antigens or living environments, or the age of the animals, and require further investigation. In summary, the results indicate that mice immunized with rTgPSA1/CS nanospheres or rTgPSA1/IFA emulsion through subcutaneous or intramuscular injection gained a significantly prolonged survival time, and there was no significant change between them.

Based on the outcomes of mice, the subcutaneous administration route attains the longest survival time, but poultry farms in reality always exist on a large scale. Thus, considering the practical reality, the intramuscular route was used in subsequent chicken trials. At commercial chicken farms, people pay more attention to the weight of chickens. Thus, the coefficient of growth was investigated after chickens were challenged with 10^7^ tachyzoites (RH strain) via intraperitoneal injection. The results indicate that a higher coefficient of growth was obtained by immunizing with two vaccines. After *T. gondii* infection, these smart parasites can disseminate widely by colonizing in leukocytes [89]. To escape immune surveillance, *T. gondii* could form cysts mainly in brain and muscle tissues [90]. Reports have shown that *T. gondii* could be detected in cardiac muscles [91,92]. To further clarify the immune protection, the parasite burden was evaluated in cardiac muscles, and a lower parasite burden was obtained after two vaccinations. All of these results demonstrate that both rTgPSA1/CS nanospheres and rTgPSA1/IFA emulsion were able to protect chickens against acute toxoplasmosis and increase their body weight.

## 5. Conclusions

This study shows that immunization with rTgPSA1/CS nanospheres and rTgPSA1/IFA emulsion could generate partial protection against acute toxoplasmosis, with a prolonged survival time in mice and a lower parasite burden in chickens. The promotion of antibodies and regulation of cytokine secretions suggest that rTgPSA1/CS nanospheres and rTgPSA1/IFA emulsion could induce strong humoral immune responses in vaccinated animals. Compared to other administration routes, the intramuscular route is optimal for CS nanospheres in mass vaccination. Furthermore, as a promising vaccine, rTgPSA1/CS nanospheres could substitute for rTgPSA1/IFA emulsion. However, the immune protection mechanisms of TgPSA1 against *T. gondii* infection in animals are complex, and further studies on this prospective vaccine should focus on its applicability to other hosts and protectivity against other *T. gondii* strains.

## Figures and Tables

**Figure 1 pharmaceutics-13-00752-f001:**
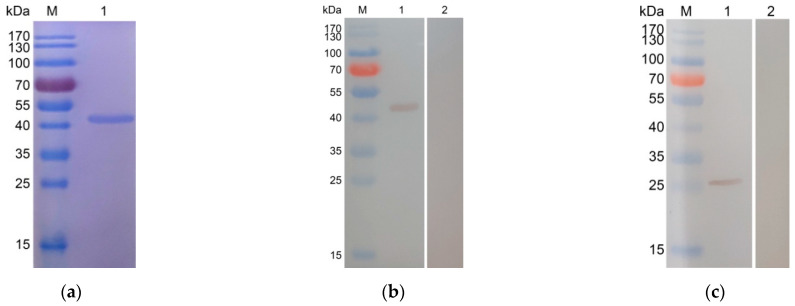
(**a**) SDS-PAGE gel of purified recombinant TgPSA1 (Line 1). Line M: molecular weight (MW) marker proteins. (**b**) Western blot analysis of recombinant TgPSA1. rTgPSA1 detected by serum from *T. gondii*-infected rats (Line 1) and healthy rats (Line 2). Line M: MW marker proteins. (**c**) Western blot analysis of native TgPSA1. Total soluble protein of *T. gondii* tachyzoites detected by sera from rTgPSA1-immunized rats (Line 1) and healthy rats (Line 2). Line M: MW marker proteins.

**Figure 2 pharmaceutics-13-00752-f002:**
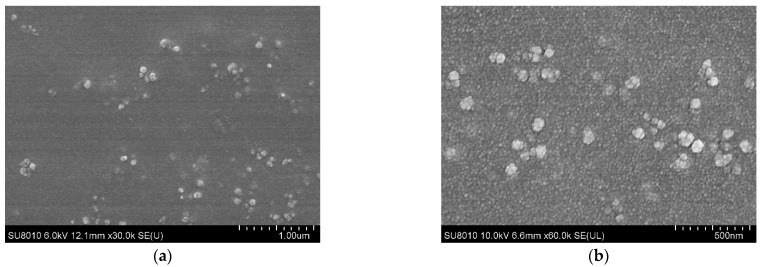
SEM images of CS nanospheres loaded with rTgPSA1 protein, showing nanosphere morphology, with bar representing (**a**) 1000 nm and (**b**) 500 nm.

**Figure 3 pharmaceutics-13-00752-f003:**
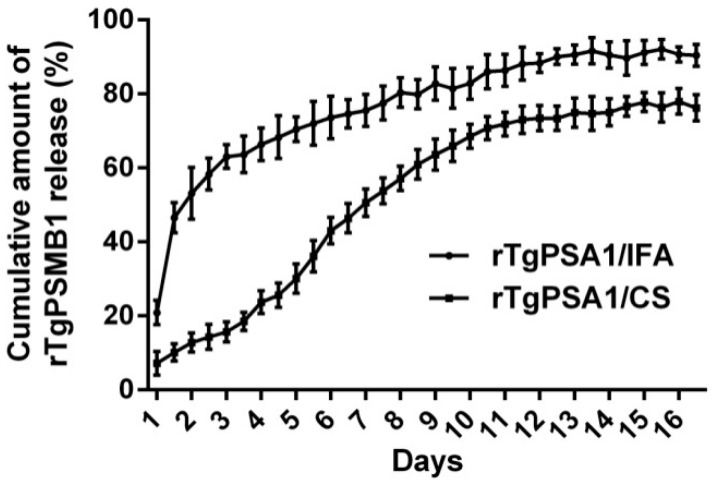
Release profile of rTgPSA1/CS nanospheres and rTgPSA1/IFA emulsion in vitro over a 16-day period. Amount of recombinant protein in supernatant was determined by BCA assay. Each sample was tested once. Values represented as mean ± SD (n = 3).

**Figure 4 pharmaceutics-13-00752-f004:**
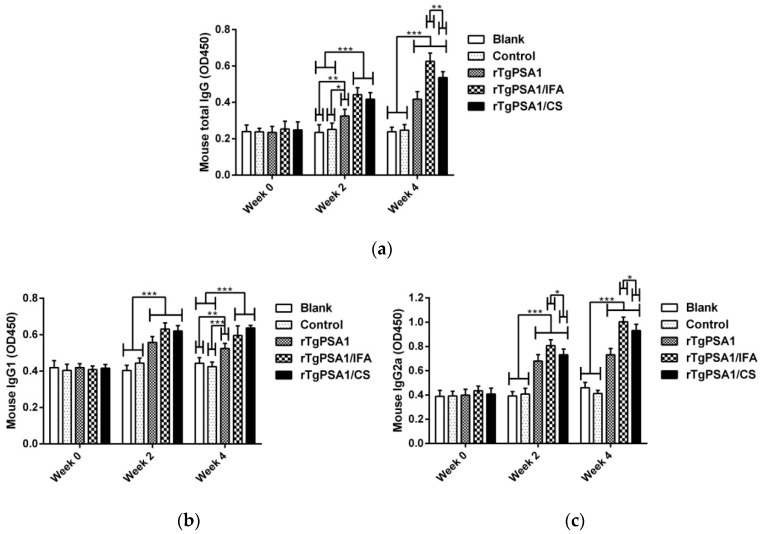
Antibody determination of (**a**) total IgG and isotypes (**b**) IgG1 and (**c**) IgG2 in sera from immunized mice at weeks zero, two, and four. Each sample was conducted once. Values were estimated using one-way ANOVA followed by Dunnett’s test and shown as mean of OD450 ± SD (n = 5). Values between rTgPSA1/IFA and rTgPSA1/CS groups were compared with the independent *t*-test. * *p* < 0.05, ** *p* < 0.01, and *** *p* < 0.001 compared with blank or control group.

**Figure 5 pharmaceutics-13-00752-f005:**
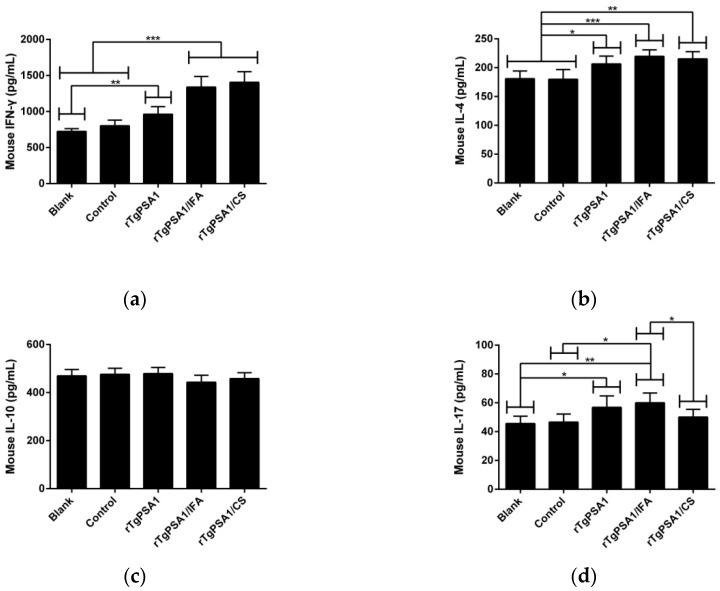
Cytokine secretions in mice. Commercially available ELISA kits were used to determine levels of (**a**) IFN-γ, (**b**) IL-4, (**c**) IL-10, and (**d**) IL-17 in mouse serum after the second immunization (week four). Each sample was conducted once. Results were estimated using one-way ANOVA tracked by Dunnett’s test, and values are shown as mean ± SD (n = 5). Values between rTgPSA1/IFA and rTgPSA1/CS groups were compared with the independent *t*-test. * *p* < 0.05, ** *p* < 0.01, and *** *p* < 0.001 compared with blank or control group.

**Figure 6 pharmaceutics-13-00752-f006:**
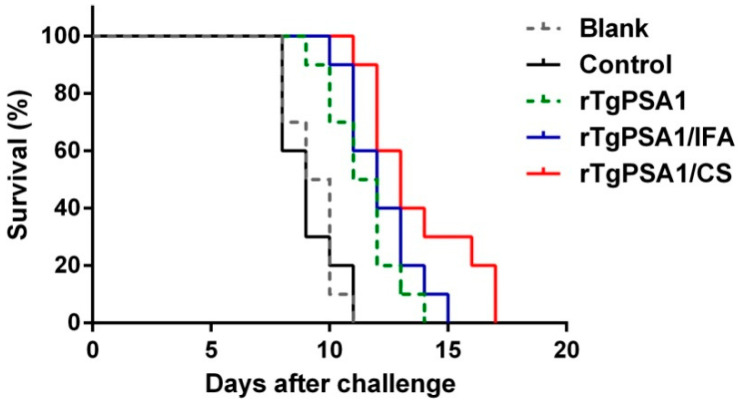
Mouse survival rates subsequently challenged by infection with 200 newly prepared tachyzoites (*T. gondii* RH strain). Results were examined by Kaplan–Meier test and survival curves were tested by log-rank model (n = 5).

**Figure 7 pharmaceutics-13-00752-f007:**
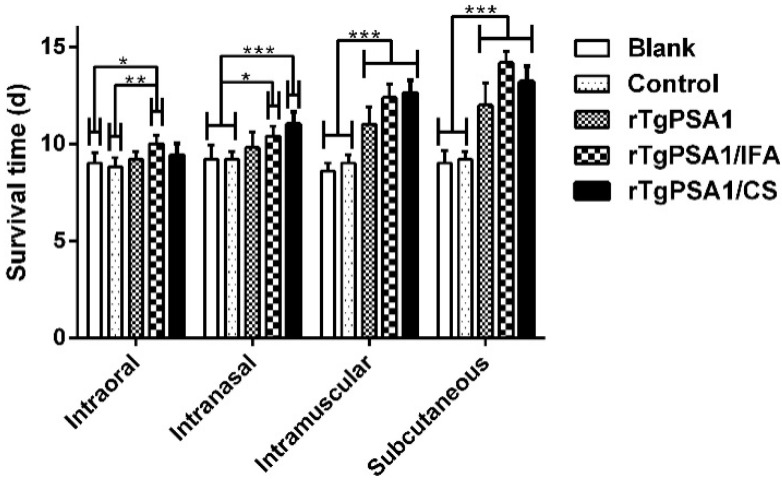
Mouse survival times with different administration routes. Two weeks after the last immunization, mice were challenged with 200 newly prepared tachyzoites (*T. gondii* RH strain) and survival time was recorded daily. Results were processed by Kaplan–Meier test and shown as mean ± SD (n = 5). Values between rTgPSA1/IFA and rTgPSA1/CS groups were compared with the independent *t*-test. * *p* < 0.05, ** *p* < 0.01, and *** *p* < 0.001 compared with blank or control group.

**Figure 8 pharmaceutics-13-00752-f008:**
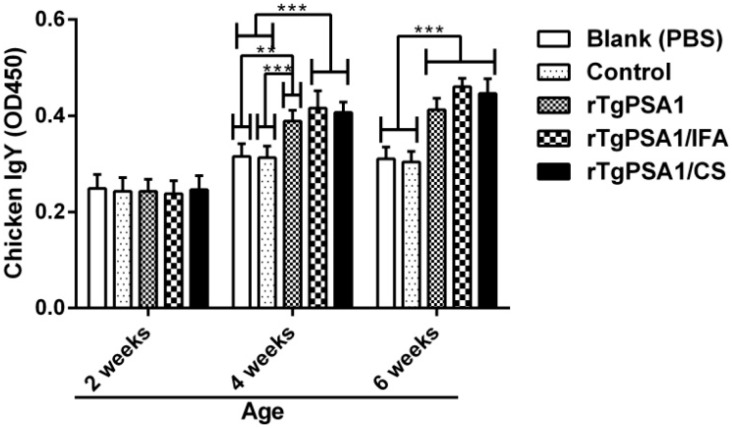
Total IgY antibody calculation in sera of chickens immunized at two, four, and six weeks of age. Each sample was conducted once. Values were assessed using one-way ANOVA followed by Dunnett’s test, and are shown as means of OD450 ± SD (n = 5). Values between rTgPSA1/IFA and rTgPSA1/CS groups were compared with the independent *t*-test. ** *p* < 0.01 and *** *p* < 0.001 compared with blank or control group.

**Figure 9 pharmaceutics-13-00752-f009:**
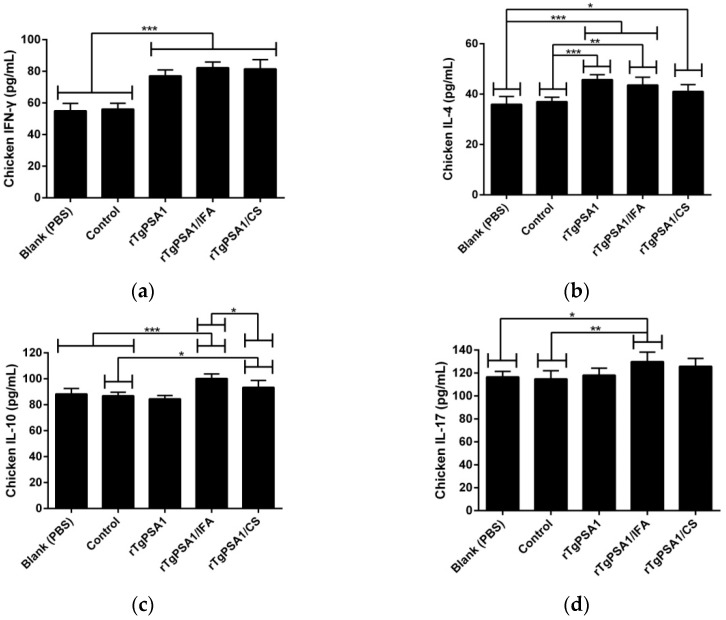
Cytokine secretions of chickens. Commercially available ELISA kits were used to determine levels of (**a**) IFN-γ, (**b**) IL-4, (**c**) IL-10, and (**d**) IL-17 in sera two weeks after second immunization (six weeks old). Each sample was conducted once. Results were evaluated using one-way ANOVA followed by Dunnett’s test, and values are shown as mean ± SD (n = 5). Values between rTgPSA1/IFA and rTgPSA1/CS groups were compared with the independent *t*-test. * *p* < 0.05, ** *p* < 0.01, and *** *p* < 0.001 compared with blank or control group.

**Figure 10 pharmaceutics-13-00752-f010:**
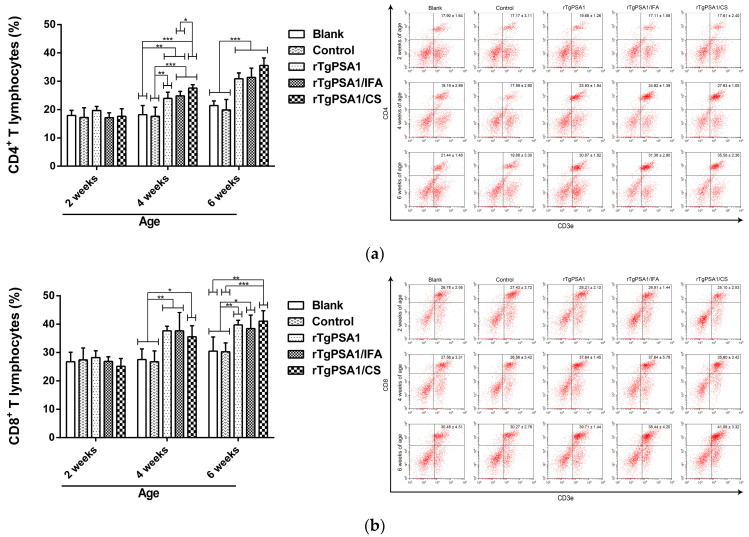
Flow cytometry analysis of (**a**) CD4^+^ and (**b**) CD8^+^ T lymphocytes in spleen lymphocytes at two, four, and six weeks of age. Five chickens in each group were sacrificed and spleen lymphocytes were collected. Each sample was conducted once. Results were estimated by one-way ANOVA followed by Dunnett’s test, and values are shown as mean ± SD (n = 5). Values between rTgPSA1/IFA and rTgPSA1/CS groups were compared with the independent *t*-test. * *p* < 0.05, ** *p* < 0.01, and *** *p* < 0.001 compared with blank or control group.

**Figure 11 pharmaceutics-13-00752-f011:**
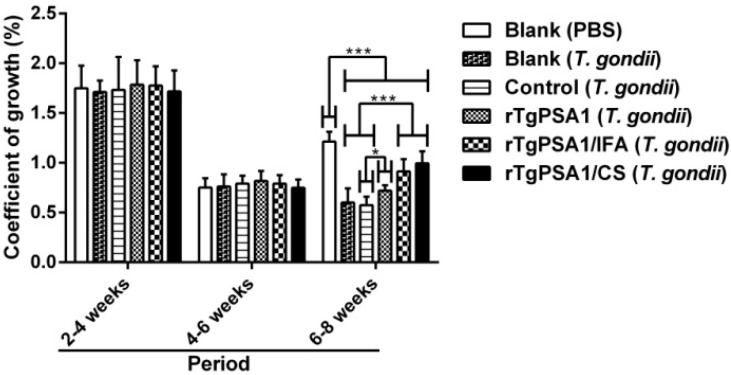
Coefficient of growth in chickens. Ten chickens were weighed at different sampling times. Results were assessed by one-way ANOVA followed by Dunnett’s test, and values are shown as mean ± SD (n = 10). Values between rTgPSA1/IFA and rTgPSA1/CS groups were compared with the independent *t*-test. * *p* < 0.05 and *** *p* < 0.001 compared with blank or control group.

**Figure 12 pharmaceutics-13-00752-f012:**
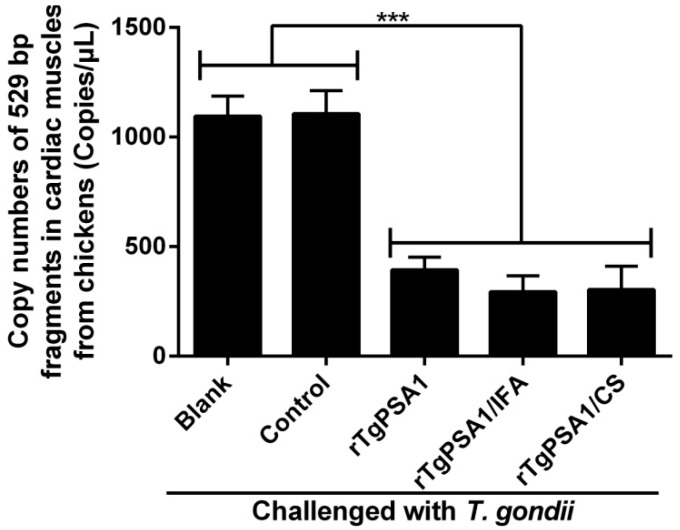
Copy numbers of 529 bp fragments in cardiac muscles from chickens. Chickens were intraperitoneally injected with 10^7^ tachyzoites of highly virulent *T. gondii* RH strain at six weeks of age (two weeks after last immunization). After another two weeks (eight weeks old), five chickens in each group were sacrificed and cardiac muscle from the tip of the heart was harvested. Each sample was run in triplicate, and the average value of each chicken was calculated. Results were evaluated using one-way ANOVA followed by Dunnett’s test, and values are shown as mean ± SD (n = 5). Values between rTgPSA1/IFA and rTgPSA1/CS groups were compared with the independent *t*-test. *** *p* < 0.001 compared with blank or control group.

**Table 1 pharmaceutics-13-00752-t001:** Immunization formulations in different mouse groups.

Group	Treatment (Each Mouse)	Time for Immunization	Time for Infection	Infection Dose (Each Mouse)
Blank	Equal volume of PBS	Week 0 and 2	Week 4	200 tachyzoites
Control	100 μg pET32a vector protein			
rTgPSA1	100 μg rTgPSA1			
rTgPSA1/IFA	rTgPSA1/IFA emulsion containing 100 μg rTgPSA1			
rTgPSA1/CS	rTgPSA1/CS nanospheres containing 100 μg rTgPSA1			
Blank	Equal volume of PBS			

**Table 2 pharmaceutics-13-00752-t002:** Immunization formulations in different chicken groups.

Group	Treatment (Each Chicken)	Time for Immunization	Time for Infection	Infection Dose (Each Chicken)
Blank (PBS)	Equal volume of PBS	At 2 and 4 weeks old	At 6 weeks old	Equal volume of PBS (0 tachyzoite)
Blank (*T. gondii*)	Equal volume of PBS			1 × 10^7^ tachyzoites
Control	200 μg pET32a vector protein			
rTgPSA1	200 μg rTgPSA1			
rTgPSA1/IFA	rTgPSA1/IFA emulsion containing 200 μg rTgPSA1			
rTgPSA1/CS	rTgPSA1/CS nanospheres containing 200 μg rTgPSA1			

## Data Availability

The data presented in this study are available within the article.

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
