# Peer review of "Toxoplasma gondii* Proteasome Subunit Alpha Type 1 with Chitosan: A Promising Alternative to Traditional Adjuvant"

_pharmaceutics, 2021, doi:10.3390/pharmaceutics13050752_

Round 1

Reviewer 1 Report

The manuscript is well written by  Yu et al. The results are clear and convincing. However, there are some issues needed to be considered for correction or modification.

Comment 1: Please try to write short and sweet title.

Comment 2: Ln.15 and Ln. 34, please remove T. gondii from bracket, scientifically its well known one.

Comment 3: Please rewrite abstract, there are lot of very good information in the results, but abstract is not that much attractive.

Comment 4: Ln. 38, its not AIDs, its AIDS, please write abbreviation.

Comment 5: Already mentioned Chitosan as CS at Ln. 80, but through out the manuscript did not follow. Some places as CS nanosphere but most of the places chitosan nanosphere. Please follow one style.

Comment 6: In materials and methods, line no. 115-142, please rewrite the section 2.2, very clearly.

Comment 7: Please dont make one or two lines as one paragraph, for example line no. 169-170; ln. no. 295-300.

Comment 8: Figure 3, repeated, please check.

Comment 9: In figure legends and through the text please follow one bond and with same size.

Comment 10: First three paragraph of discussion section, looks like introduction, please rewrite, lot of results are there please compare your results with recently published papers.

Comment 11: Please rewrite lost two lines of conclusions, if possible rewrite conclusion.

Comment 12: Please check English and Grammar through out the manuscript. Use appropriate scientific words in some places. Really appreciate your efforts.

Reviewer 2 Report

The paper “Chitosan nanospheres entrapped recombinant proteasome subunit alpha type 1 of Toxoplasma gondii resist toxoplasmosis in mice and chickens” is an exhaustive study of the use of chitosan in the development of an efficient vaccine against acute toxoplasmosis. The study was well conducted, the methodology was well chosen and the findings are sound. I recommend its publication, after some minor revisions, as follows.

  1. The use of chitosan in the development of systems against toxoplamosis should be discussed in the Introduction.
  2. The “Preparation and physical characterization of chitosan nanospheres loaded with rTgPSA1” will be carefully revised in order to avoid the telegraphic writing manner, to be clearer for the readers.
  3. Along the manuscript, carefully highlight the benefits of using chitosan compared to Incomplete Freund's adjuvant.
  4. Carefully revise the English language, here and there, there are sentences without meaning, e.g. in Conclusions: “In summary, as a promising material for use as substitute to incomplete Freund's adjuvant. Chitosan nanospheres loaded with rTgPSA1 could be a promising method to develop a vaccine against toxoplasmosis.”

Round 2

Reviewer 1 Report

Comment 1: Line no. 20, please change chitosan nanosphere to CS nanosphere. Comment 2: Line no. 21, 22; in vitro; in vivo change in to italics Comment 3: Please mention PCR conditions in section 2.2 Comment 4: Please check Figure 3, repeated.

Reviewer 2 Report

The paper has been properly revised and it is suitable for publication now.
